# Minimal versus specialist equipment in the delivery of pulmonary rehabilitation: protocol for a non-inferiority randomised controlled trial

Claire M Nolan [ID] ,[1,2] Jessica A Walsh,[1] Suhani Patel,[1,2] Ruth E Barker,[1,2] Oliver Polgar,[1] Matthew Maddocks,[3] Wei Gao [ID] ,[3] Rebecca Wilson,[3] Francesca Fiorentino,[4,5] William Man[1,2,6]

For numbered affiliations see end of article.

**Correspondence to**
Dr Claire M Nolan;
c.nolan@rbht.nhs.uk

## ABSTRACT

**Introduction** Pulmonary rehabilitation (PR), an exercise and education programme for people with chronic lung disease, aims to improve exercise capacity, breathlessness and quality of life. Most evidence to support PR is from trials that use specialist exercise equipment, for example, treadmills (PR-gym). However, a significant proportion of programmes do not have access to specialist equipment with training completed with minimal exercise equipment (PR-min). There is a paucity of robust literature examining the efficacy of supervised, centre-based PR-min. We aim to determine whether an 8-week supervised, centre-based PR-min programme is non-inferior to a standard 8-week supervised, centre-based PR-gym programme in terms of exercise capacity and health outcomes for patients with chronic lung disease.

**Methods and analysis** Parallel, two-group, assessor-blinded and statistician-blinded, non-inferiority randomised trial. 436 participants will be randomised using minimisation at the individual level with a 1:1 allocation to PR-min (intervention) or PR-gym (control). Assessment will take place pre-PR (visit 1), post-PR (visit 2) and 12 months following visit 1 (visit 3). Exercise capacity (incremental shuttle walk test), dyspnoea (Chronic Respiratory Questionnaire (CRQ)-Dyspnoea), health-related quality of life (CRQ), frailty (Short Physical Performance Battery), muscle strength (isometric quadriceps maximum voluntary contraction), patient satisfaction (Global Rating of Change Questionnaire), health economic as well as safety and trial process data will be measured. The primary outcome is change in exercise capacity between visit 1 and visit 2. Two sample t-tests on an intention to treat basis will be used to estimate the difference in mean primary and secondary outcomes between patients randomised to PR-gym and PR-min.

**Ethics and dissemination** London-Camden and Kings Cross Research Ethics Committee and Health Research Authority have approved the study (18/LO/0315). Results will be submitted for publication in peer-reviewed journals, presented at international conferences, disseminated through social media, patient and public routes and directly shared with stakeholders.

**Trial registration number** ISRCTN16196765.

## Strengths and limitations of this study

► This is a parallel, two-group, assessor-blinded and statistician-blinded, non-inferiority randomised trial to assess the primary and secondary outcome measures.

► The pulmonary rehabilitation intervention is delivered according to national guidelines: British Thoracic Society Guideline on Pulmonary Rehabilitation in Adults.

► The study includes a 12-month follow-up period to evaluate the long-term effect of the intervention.

► The study includes a health-economic analysis which aims to understand relative cost-effectiveness which may have implications for NHS commissioning.

► This is a single-centre study based in northwest London and findings may not be applicable to other populations in different locations.

## INTRODUCTION
### Background and rationale
Pulmonary rehabilitation (PR), an evidence-based exercise and education programme, is widely accepted as a cornerstone of management for people with chronic obstructive pulmonary disease (COPD) and other chronic respiratory disorders.[1] The most recent Cochrane review, comprising 65 randomised controlled trials (n=3822), stated that no further trials comparing PR and standard care in patient with COPD were necessary as the evidence supporting the benefits of PR on exercise capacity and health-related quality of life were conclusive.[2]

The majority of evidence to support PR has come from trials conducted in hospital or rehabilitation centres that use specialist exercise equipment such as treadmills, cycle ergometers and weights machines (PR-gym).[2] However, in clinical practice, supply does

not meet demand, and routine access to specialist equipment may not be feasible.[3] For example, the 2015 Royal College of Physicians National Audit identified that PR services in England and Wales received 68 000 referrals in 2014 out of an estimated 446 000 eligible patients with COPD.[3] Furthermore, 82% of patients enrolled in a PR programme in England and Wales in 2017 were hosted in a community site such as a community centre or church hall, of which 61% most likely did not exercise using specialist equipment.[4] Accordingly, exercise training at these sites, although supervised by PR professionals, was completed with minimal equipment (PR-min), typically using portable equipment such as free weights, elastic resistance bands (eg, Theraband), walking courses and bodyweight resistance exercises.[3]

Apart from improving accessibility, it has been argued that PR-min may have other advantages over PR-gym. Exercise-training using minimal equipment may better reflect activities of daily-living than training using specialist equipment and therefore be easier to replicate and maintain at home following discharge from PR. However, it may be more difficult to prescribe and progress exercise, particularly resistance exercise, without the use of specialist equipment.

A systematic review, conducted by Alison and colleagues, identified eight randomised controlled trials that compared outcomes following exercise interventions completed with minimal equipment to usual care without an exercise intervention in patients with COPD.[5] The results were conflicting for exercise outcomes. In four studies (n=182) where the 6 min walk test was used as the primary outcome measure of exercise capacity, the pooled effect showed a mean (95% CI) between-group difference of 40 (13 to 67) m favouring the intervention group.[5] Conversely, in the other four studies (n=389) that used the incremental shuttle walk test (ISW), there was no significant mean (95% CI) between-group difference: 21 (−9 to 51) m.[5] For health-related quality of life, results were similarly conflicting.[5] Whereas in the four studies that used the St. George's Respiratory Questionnaire there was a sizeable difference between the intervention and control groups (mean (95% CI): −7 (−12 to −3)), the mean between-group difference of the Chronic Respiratory Questionnaire dyspnoea (CRQ-D) and fatigue domains (CRQ-F) did not reach the minimum clinically important difference (MCID) in three other studies.[5] Only one of the eight studies measured muscle strength but results were reported descriptively with no statistical testing.[5] Furthermore, only one of the eight studies would fulfil the British Thoracic Society (BTS) guidelines definition of PR,[1] with seven studies not offering any education component.[5]

To our knowledge, no trial has compared supervised centre-based PR using minimal and specialist equipment. A previous National Institute for Health Research Health Technology Assessment-funded randomised 2×2 trial compared PR undertaken in community venues with PR undertaken in a hospital venue, in 240 patients with COPD.[6] Participants were block randomised (hospital n=129; community n=111). Both groups received two times a week PR for 6 weeks with the exercise training protocol identical in both venues. Importantly, neither community nor hospital sites had access to specialist exercise equipment. Patients in both groups improved their walking distance, exceeding the MCID of the endurance shuttle walk test, and there were similar improvements in health-related quality of life.[6] Although the results support the conclusion that the efficacy of PR-min is not influenced by hospital or community locations, this study did not address whether PR-min is non-inferior to 'gold standard' PR-gymOur group recently conducted an observational cohort study to compare if an 8 week supervised, centre-based PR-min programme was non-inferior to PR-gym in terms of core PR outcomes in patients with COPD.[7] Using propensity score matching, 318 consecutive patients undergoing PR-min were compared 1:1 with a control group of 318 patients who underwent PR-gym. Similar short-term improvements in exercise capacity and health-related quality of life were observed in both groups (mean (95% CI) difference: ISW: 3 (−16 to 9) m; CRQ-total: 0.9 (−2.7 to 4.5)). Furthermore, the 95% CI between-group differences for these outcomes did not cross the predefined non-inferiority margins. These data suggest that PR-min is non-inferior to PR-gym with regard to short-term improvements in exercise capacity and health-related quality of life. However, this study was not randomised, only included short-term outcomes and no attempt was made to estimate costs. A randomised controlled trial with more varied outcome measures (such as lower limb muscle strength), longer follow-up and a health economic analysis, is required to confirm these findings.

The aims of the research are to determine whether an 8-week outpatient supervised PR-min programme is non-inferior to a standard 8-week outpatient supervised PR-gym programme in terms of health benefits for patients with chronic respiratory disease.

The primary objective is to determine whether PR-min is non-inferior to PR-gym regarding change in exercise capacity measured using ISW[8] distance from baseline (visit 1) to post-PR assessment at 8 weeks (visit 2).

The secondary objectives are to: (1) determine whether PR-min is non-inferior to PR-gym regarding changes breathlessness (CRQ-D), health-related quality of life (CRQ) and quadriceps strength (isometric quadriceps maximum voluntary contraction) from baseline (visit 1) to post-PR assessment at 8 weeks (visit 2) and from baseline (visit 1) to 12 months after the baseline assessment (visit 3); (2) evaluate the trial process by recording the participant recruitment and retention, participant uptake of PR, PR attendance, PR completion, reasons for PR non-completion and participant satisfaction in each arm of the study, at the appropriate stage of the trial; (3) estimate the cost and cost-effectiveness during the trial period.

## METHODS AND ANALYSIS

### Ethical approval

This study has been approved by the Health Research Authority and London Camden and Kings Cross Research Ethics Committee (reference 18/LO/0315). All participants will provide written informed consent.

### Study design

This is a parallel, two-group, assessor-blinded and statistician-blinded, non-inferiority, randomised trial. Participants will be randomised at the individual level with a 1:1 allocation to either PR-min or PR-gym. Both interventions will comprise two supervised sessions per week for 8 weeks delivered by the same team. Outcome measures will be recorded at baseline assessment for PR (visit 1), following PR at 8 weeks (visit 2) and at 12 months after visit 1 (visit 3). The study schema is outlined in figure 1.

### Study population

Potential participants will be people referred to Harefield Pulmonary Rehabilitation Unit, Royal Brompton and Harefield Clinical Group, UK who meet the following inclusion criteria (1) adults >18 years of age; (2) physician diagnosis of stable chronic respiratory disease, typically COPD, interstitial lung disease, bronchiectasis, chronic asthma or chest wall disease; (3) referred for PR in line with BTS guidelines (ie, ambulatory—can walk ≥5

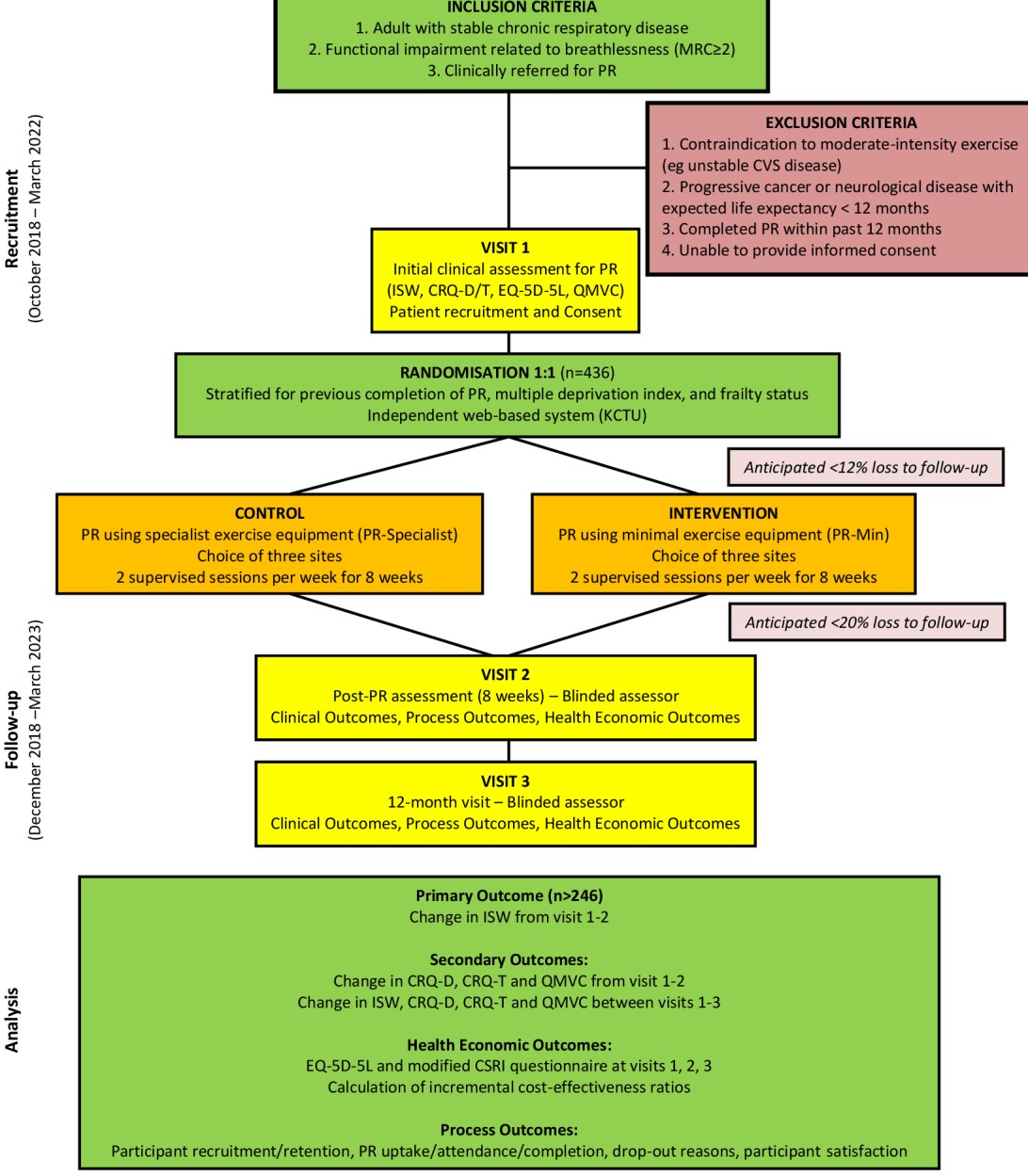

**Figure 1** Study schema. CRQ-D, Chronic Respiratory Questionnaire-Dyspnoea; CRQ-T, Chronic Respiratory Questionnaire-Total Score; CVS, Cardiovascular Disease; ISW, Incremental Shuttle Walk Test; KCTU, King's Clinical Trials Unit; MRC, Medical Research Council; PR, Pulmonary Rehabilitation; QMVC, Quadriceps Maximum Voluntary Contraction.

m, functional impairment related to breathlessness, typically Medical Research Council dyspnoea score ≥2); (4) able to communicate verbally and respond to questions in written English. Potential participants will be excluded if they have (1) any contraindication to moderate intensity physical exercise, for example, unstable cardiovascular disease; (2) progressive cancer or neurological disorder with an expected life expectancy less than 12 months; (3) completed PR within previous 12 months and/or (4) are unable to provide informed consent.

### Consent

Informed consent will be obtained by the nominated researchers as recorded in the Sponsor's Delegation of Responsibilities Log. All individuals taking informed consent will have received consent training. All participants will provide written consent. Consent to enter this study will be obtained after a full account has been provided of its nature, purpose, risks, burdens and potential benefits, and the patient has had the opportunity to deliberate. The patient will be allowed to specify the time they wish to spend deliberating, usually up to 24 hours. The Investigator or designee will explain that the patients are under no obligation to enter the study and that they can withdraw at any time during the study, without having to give a reason. At each visit, the participant's willingness to continue in the study will be ascertained and documented in the research notes. A copy of the consent form is in the online supplemental file.

### Screening and recruitment

Following clinical referral, but prior to pre-PR assessment, potential participants will be screened by a member of the PR team for eligibility to the study. The potential participant will be approached by telephone and provided with a participant information sheet. A member of the research team will obtain informed consent following the pre-PR assessment.

### Randomisation, allocation and blinding procedures

Consenting participants will be randomised at the individual level with a 1:1 allocation, using an independent web-based system provided by the United Kingdom Clinical Research Collaboration registered King's Clinical Trials Unit (KCTU), to receive either 'usual care' (PR-gym) or intervention (PR-min).

Randomisation by minimisation will ensure that participant characteristics are balanced between the groups with respect to previous completion of PR (yes/no), multiple deprivation index (most deprived quintile of index: yes/no[9] and frailty status (Short Physical Performance Battery score <10/≥10.[10] A proportion of patients will be entered initially using simple randomisation in order to create a level of initial imbalance and the minimisation algorithm will maintain a level of randomness in order to preserve prerandomisation allocation concealment. Once randomised, the system will automatically generate a full audit trail of the process and send email to

relevant investigators in a blinded or unblinded format, depending on their role. For each arm, participants will be provided with a choice of three sites in northwest London to undertake PR.

Owing to the nature of the interventions, participants and providers of the intervention will not be blinded. However postrehabilitation assessments will be performed by a researcher blinded to group allocation and not involved in the delivery of either intervention arm. The trial statistician will also be blinded to group allocation.

### Study intervention

#### PR-gym (control group)

The control intervention will be current gold-standard clinical practice. PR-gym will comprise an 8-week outpatient exercise and multidisciplinary self-management education programme, with two supervised and at least one additional home session each week, and delivered according to the BTS Quality Standards for PR.[11] Supervising staff will comprise specialist respiratory therapists with a minimum of 2 years' experience in PR. Available equipment will include treadmills, cycle ergometers, cross-trainers, specialist lower limb resistance equipment (eg, leg press, knee extension). Each supervised session will consist of 1 hour of exercise (at least 30 min aerobic exercise) and 45 min of education.

Initial walking speed prescription on the treadmill will be 80% of predicted peak oxygen consumption based on baseline ISW performance,[8] while initial endurance cycling will be initially set to achieve level 3 to 4 on the Borg CR-10 Dyspnoea Scale with the aim of patients completing 10 min of continuous training. Lower limb resistance training will comprise two sets of 10 leg press repetitions on specialist resistance equipment performed with an initial training load of 60% one-repetition maximum. Similarly, two sets of 10 bilateral knee extension repetitions will be performed on specialist resistance equipment at an initial training load of 60% one-repetition maximum. This will be supplemented with sit-to-stand sets, plus knee lifts/extension and hip abduction with appropriate ankle weights up to 10 kg. Upper limb resistance training will comprise biceps curls, shoulder press and upright row with free weights or Therabands (red to black). Exercise training will be individualised and regularly progressed (duration and/or intensity) according to standard operating procedures, with targets reviewed at each session. Education will be delivered by a multidisciplinary team with topics chosen to develop patients' understanding and holistic management of their disease. Further detail on the education programme and safety measures for the rehabilitation programme are provided in the online supplemental file.

#### PR-min (Intervention group)

PR-min will also comprise an 8-week outpatient exercise and multidisciplinary self-management education programme, with two supervised and at least one additional home session each week, and delivered according

to the BTS Quality Standards for PR.[11] As for PR-gym, supervising staff will comprise specialist respiratory therapists with at least 2 years' experience of in PR. There will be no access to treadmills, cycle ergometers or specialist resistance equipment. Available exercise equipment will include walking circuit, portable steppers, portable pedals, hand and ankle weights (up to 5 kg), and Therabands (red to black). Each supervised session will consist of 1 hour of exercise (at least 30 min aerobic exercise) and 45 min of education.

Initial walking speed prescription will be 80% of predicted peak oxygen consumption based on baseline ISW performance. Participants will be provided with a stopwatch and given time targets to complete a walking course of known distance. Although the resistance of the portable steppers and pedals can be manually adjusted, this cannot be objectively quantified. Initial prescription will be set at 'level 1' but individually adjusted to find an intensity where patients can complete 10 min of continuous training with a target modified Borg breathless score of 3–4 and a Borg rating of Perceived Exertion of 13–15 (on a scale of 6–20). Resistance training will include functional activities such as sit-to-stand and step-ups as well as Theraband based exercises such as sitting knee extension, leg press and hip flexion as well as standing hip extension, squats, chest press and lateral raise. Progression will be through the use of hand/ankle weights and increasing resistant Therabands (from red to black). Exercise training will be individualised and regularly progressed (either in duration or intensity) with targets reviewed at each supervised session. Education will be delivered by a multidisciplinary team as per PR-gym (online supplemental file). Information on safety measures for the rehabilitation programme is provided in the online supplemental file.

## Study outcome measures

A list of the scheduled outcomes is outlined in table 1 and described in detail in the online supplemental file.

## Sample size calculation

Previous audits of Harefield PR Unit have shown that in participants undergoing PR-gym achieve a mean (SD) change in ISW of 58 (67) m. The null hypothesis is that PR-min is inferior to the standard treatment PR-gym. The alternative hypothesis is that PR-min is not inferior to PR-gym. The non-inferiority margin will be defined as half the known MCID using the fixed-margin method with a preserved effect of 50% as recommended by previous guidance, including from the United States Food and Drug Administration.[12 13] The MCID of the ISW is 47.5 m,[14] and therefore 24 m will be considered the non-inferiority margin. If there is truly no difference between PR-min and PR-gym, then a minimum of 246 patients (123 in each group) is required to be 80% sure that the lower limit of a one-sided 97.5% CI (or equivalently a 95% two-sided CI) will be above the non-inferiority limit of –24 m. Based on audit data, we anticipate 32% dropout from PR (12% from assessment to starting PR and 20% from starting PR to completing PR). Taking into account dropout, the original minimum sample size required for analysis was 362 patients (181 patients per group). However, owing to the impact of the COVID-19 pandemic on the study (described in the Amendment section), the study sample size was increased by a further 74–436 (218 patients per group).

## Statistical analysis

For baseline data, continuous and categorical variables will be summarised with descriptive statistics. No significance testing will be carried out. The outcome measures will be described by trial arm and at visits 1, 2 and 3 using descriptive statistics.

The main statistical analyses will estimate the difference in mean primary and secondary outcomes between patients randomised to PR-gym and PR-min by intention to treat principle from visit 1 to visit 2 (8 weeks following visit 1) and visit 1 to visit 3 (12 months following visit 1). Group difference estimates and associated one-sided 97.5% CI will be reported. The group differences will be compared using one-sided two independent sample t-test

| Table 1 Study outcome measures | | | |
|---|---|---|---|
| | Visit 1 | Visit 2 | Visit 3 |
| Primary outcome measure: exercise capacity (incremental shuttle walk test) | X | X | X |
| Spirometry | X | | X |
| Frailty (Short Physical Performance Battery) | X | X | X |
| Dyspnoea (Chronic Respiratory Questionnaire-Dyspnoea domain) | X | X | X |
| Health-related quality of life (Chronic Respiratory Questionnaire) | X | X | X |
| Muscle strength (Isometric Quadriceps Maximum Voluntary Contraction) | X | X | X |
| Patient satisfaction (Global Rating of Change Questionnaire) | | X | |
| Health-economic evaluation (Modified Client Service Receipt Inventory Scale, Euro-Qol-5 Dimensions-5 Levels, NHS Digital—after visit 3 only) | X | X | X |
| Safety and trial process evaluation | X | X | X |

or non-parametric equivalent. The significance level is set at one-sided significance level of 0.025.

Regarding missing data, the number of baseline variables with complete data will be reported and the primary and secondary outcomes will be analysed as per the recommendations of White *et al.*[15] Missing postrandomisation assessments will be dealt with using an appropriate method of imputation, depending on the missing mechanism.

Adherence is defined as the number of supervised sessions the participant attends irrespective of the assigned intervention (maximum 16). This will be recorded in both intervention arms. As a dichotomous indicator with eight as the cut-off. The following variables will be recorded in a 3×2 table according to allocated intervention: (1) compliers always receive the allocated treatment; (2) complete defiers do the opposite of the allocated treatment, that is, attend no sessions in allocated treatment; (3) partial defiers are all others not falling in the above groups. Dropout cause will be recorded using the MORECARE classification of reason for attrition.[16]

In addition to the primary intention-to-treat analysis, the effect of actually receiving treatment as defined in the protocol will also be estimated. If non-compliance rate with PR sessions is >10%, a Complier-Average Causal-Effect (CACE) will be estimated.[17] The following sensitivity analyses are planned: (1) per protocol; (2) including only those in the upper quartile for baseline ISW; (3) complete case; (4) complier Average Causal Effects (CACE) analysis[17] and (5) primary diagnosis of COPD; (6) a generalised estimating equation-based analysis to estimate the treatment effects, adjusting for imbalance if the group difference in the percentage of smokers, participants age <70 years, or females is greater than 20% at baseline.

### Data collection, management and monitoring
A web based electronic data capture (EDC) system has been designed using the InferMed Macro 4 system and will be maintained by KCTU for the duration of the project. Source data will be entered by recruiting site staff, typically within 1 week of data collection by authorised staff onto the EDC. A full audit trial of data entry and any subsequent changes to entered data will be automatically date and time stamped, alongside information about the user making the entry/changes within the system.

The primary investigator (WM) will take overall responsibility for the conduct of MISTER. The trial co-ordinator (CMN) will supervise the day-to-day operation of the project and is responsible for ensuring that Good Clinical Practice guidelines are followed. An unblinded member of the MISTER research team (JAW) will monitor the data and review a random sample of 10% of completed case report forms against clinical records. Monitoring will ensure protocol compliance, proper study management and timely completion of study procedures.

### Monitoring and auditing
A Data Monitoring and Ethics Committee has been appointed. The members of this committee are not members of the applicants' or sponsors' institution and include an academic Consultant Respiratory Physician and an independent statistician.

The requirement for study monitoring or audit will be based on the Sponsor's Research Office risk assessment procedure and applicable standard operating procedures. It is the responsibility of the Research Office to determine the monitoring risk assessment and explain the rationale to the study research team.

### Patient and public involvement (PPI)
The concept of the research project arose directly from PPI, and the development of this project has included PPI throughout each stage. We have a named PPI coapplicant who will continue to be closely involved in the design and management of the research through membership of the Trial Steering Group that will meet at the beginning of the research period, and then every 6 months throughout the data collection/analysis and dissemination period. This PPI member will be joined by a second representative and will meet the project manager at regular intervals throughout the study. The PPI group will provide input into written material for patients, how results data are presented, particularly to lay audiences and will have a role in dissemination of research findings.

### Safety monitoring
A serious adverse event (SAE) is defined as an untoward occurrence that (a) results in death; (b) is life-threatening; (c) requires hospitalisation or prolongation of existing hospitalisation; (d) results in persistent or significant disability or incapacity or (e) is otherwise considered medically significant by the investigator.

The Investigator and research team are responsible for reporting events to the Research Office immediately and/or within 24 hours of becoming aware of the event. An SAE occurring to a research participant will be reported to the Research Ethics Committee that gave a favourable opinion of the study.

### Amendment
Owing to the COVID-19 pandemic the study was suspended for 13 months between February 2020 and 2021 and, at the date of study suspension, there were 74 recruited and randomised participants waiting to start PR but were unable to receive either intervention due to the pandemic. Accordingly, a substantial amendment to increase the sample size from 362 to 436 and extend the recruitment and follow-up periods by 1 year to 31 March 2022 and 2023, respectively, was submitted to the funders (National Institute for Health Research), the Health Research Authority and research ethics committee on 5 February 2021. These amendments were approved on 24 February 2021.

Should further substantial amendments be made, a notice of amendment will be submitted to the Health Research Authority for consideration.

## Study timelines

Participant recruitment started on 15 October 2018 and is expected to end on 31 March 2022. Visit 2 assessments started approximately 8 weeks after visit 1 and visit 3 assessments 12 months after visit 1. The final visit 2 assessment is expected to take place on 31 May 2023. The end date for the study, representing the final visit 3 will be 31 March 2023.

## DISSEMINATION

We will use a broad strategy to maximise dissemination of our findings: (1) sharing of scientific findings via open-access publication in high impact journals and presentation at international meetings; (2) plain English summaries of findings for public bodies and web-sites (eg, NIHR, CLARHC, British Lung Foundation, patient. co.uk) to communicate evidence through a user-friendly interface; (3) online pages about the project on websites of contributing organisations (Guy's and St Thomas' NHS Foundation Trust, Cicely Saunders Institute, King's College London); (4) social media (eg, Twitter); (5) public engagement via talks with service user groups and open public events, so patients and their caregivers can learn about the research; (6) direct sharing of findings with public bodies (eg, National Horizon Scanning Research and Intelligence Centre) and policy makers to whom the applicants have direct access (eg, BTS Pulmonary Rehabilitation Advisory Group, European Respiratory Society, American Thoracic Society, Society for Research in Rehabilitation) to facilitate uptake of findings into research strategy and policy; (7) education of commissioners to influence future NHS service delivery.

## DISCUSSION

PR is a core strategy in the management for COPD and other chronic respiratory disorders.[1] The majority of evidence to support PR has come from trials conducted in locations that use specialist exercise equipment.[2] Data from a national audit indicate that many PR programmes in England and Wales do not have access to specialist exercise equipment with exercise training completed using simple equipment.[3] There is a paucity of robust literature examining the efficacy of PR-min and to our knowledge, there have been no trials comparing supervised centre-based PR-min and PR-gym. One observational study demonstrated that PR-min is non-inferior to PR-gym in terms of exercise capacity and health-related quality of life but indicated that further investigation using a randomised controlled trial is required.[7] Therefore, this study aims to bridge the gap in knowledge by identifying whether PR-min is non-inferior to PR-gym regarding change in core PR outcome measures. This is an important and current health service question given the increasing disparity between supply and demand of PR that has implications for the delivery and future development of this programme at national and international levels.

The strengths of this study include the design: a parallel, two-group, assessor-blinded and statistician-blinded, non-inferiority randomised trial which is the appropriate design to assess the primary and secondary outcome measures. The intervention, PR, is delivered according to national guidelines and the population of interest is people with chronic respiratory disease, both of which mean the results will be generalisable. In addition, study involves a 12-month follow-up period (from visit 1) to evaluate the long-term effect of the intervention and a health-economic analysis which aims to understand the potential cost-effectiveness of the trial which may have implications for the NHS and Clinical Commissioning Groups. A limitation of the study is that it is a single-centre study based in northwest London which may mean that the findings may not be generalisable to other populations in different locations.

**Author affiliations**
[1]Harefield Respiratory Research Group, Guy's and St Thomas' Hospitals NHS Trust, London, UK
[2]National Heart and Lung Institute, Imperial College London, London, UK
[3]Cicely Saunders Institute, Division of Palliative Care, Policy & Rehabilitation, King's College London, London, UK
[4]Imperial Clinical Trials Unit, Imperial College London, London, UK
[5]Department of Surgery and Cancer, Imperial College London, London, UK
[6]Harefield Pulmonary Rehabilitation Unit, Royal Brompton and Harefield Clinical Group, Guy's and St Thomas' NHS Foundation Trust, London, UK

**Acknowledgements** We would like to acknowledge Joanna Kelly, Evangelos Georgio, Janice Jimenez and their teams in the Kings Clinical Trials Unit data centre who developed and will maintain the randomisation and Elsevier MACRO database systems for the trial. We would also like to acknowledge Ms Peihan Yu, statistician at Kings College London, for her help developing the study protocol. We would like to express our gratitude to the Harefield Pulmonary Rehabilitation Unit, Royal Brompton and Harefield Clinical Group, Guy's and St Thomas' NHS Foundation Trust who will help with participant identification and delivery of the pulmonary rehabilitation programme.

**Contributors** CMN, MM, WG and WM: concept and design of the study. CMN, JAW, SP, REB, OP: data collection and entry. CMN, MM, WG and WM: drafted the manuscript. WG, RW, MM, CMN and WM: drafted the statistical analysis plan. All authors reviewed the final draft of the manuscript.

**Funding** This work is supported by National Institute for Health Research grant number PB-PG-0816–20022.

**Competing interests** CMN reports fees from Novartis outside of this research. MM consultancy fees from Fresenius Kabi and Helsinn, grants from National Institute for Health Research, grants from British Lung Foundation, outside the submitted work. WM reports personal fees from Jazz Pharmaceuticals, personal fees from Mundipharma, personal fees from Novartis, grants from Pfizer, non-financial support from GSK, grants from National Institute for Health Research, grants from British Lung Foundation, outside the submitted work.

**Patient consent for publication** Not applicable.

**Provenance and peer review** Not commissioned; externally peer reviewed.

responsibility arising from any reliance placed on the content. Where the content includes any translated material, BMJ does not warrant the accuracy and reliability of the translations (including but not limited to local regulations, clinical guidelines, terminology, drug names and drug dosages), and is not responsible for any error and/or omissions arising from translation and adaptation or otherwise.

**ORCID iDs**
Claire M Nolan http://orcid.org/0000-0001-9067-599X
Wei Gao http://orcid.org/0000-0001-8298-3415

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
