## [Reviewer comments · BMJ Open]

ARTICLE DETAILS

TITLE (PROVISIONAL)	Minimal versus specialist equipment in the delivery of pulmonary rehabilitation: Protocol for a non-inferiority randomised controlled trial
AUTHORS	Nolan, Claire; Walsh, Jessica; Patel, Suhani; Barker, Ruth; Polgar, Oliver; Maddocks, Matthew; Gao, Wei; Wilson, Rebecca; Man, William

VERSION 1 – REVIEW

REVIEWER	Heine, Martin Stellenbosch University, Institute of Sports and Exercise Medicine
REVIEW RETURNED	13-Jan-2021

GENERAL COMMENTS	Dear Dr. Nolan and colleagues, Please accept my compliments for the design and write-up of this exciting study on the non-inferiority of minimal-equipment pulmonary rehabilitation versus gym-based PR. The study design is well written, easy to follow, well thought out, robust, and highly relevant in light of the increasing NCD pressure on the health care system (whether in the UK, or beyond). After carefully reading the submitted protocol and online supplement (comments in attached file), I've recommended minor revisions. I wish you all the best in this huge undertaking (in particular in current times), and would be more in happy to receive feedback on how the study is progressing. Working in a South African setting, we often are faced with many and diverse resource constraints, of which the availability of equipment is often one of them. Hence, I'm personally interested in the outcome of your work. Best wishes, Martin Heine, PhD Main comments Page 7; Methods - There are no timelines provided for the study (e.g. start of recruitment etc, and anticipated duration of the study). Please include. Page 9 – 11; Interventions - While I understand this is unlikely at this phase of the study, it would have been great to have a third arm in which PR-min was
---

	provided virtually / remotely, to an home-based environment. This would expand the external validity of the study to settings where also the physical resources, human-resources, and/or personal context does not facilitate outpatient visits (beyond the implications of COVID-19).  - Please add detail on any interaction between participant and team during, or in preparation of, the follow-up period. Are those in PR-min provided with the tools to continue their program, are patients in PR-gym directed to appropriate facilities in their home environment? Will there be phone follow-ups, booster sessions etc? - The “education component” should be expanded upon, as this comprises a sizeable portion of the intervention and could be an important driver for changes in health-related quality of life (directly), as well as exercise-related outcomes (indirectly, through promotion of disease understanding and continued exercise during follow-up phase specifically). Page 10, line 10  - Please consider adding these SOPs (progression schedule) as appendix Page 12, statistical analysis  - Would a more contemporary analysis like multi-level or GEE not be more appropriate given the longitudinal nature of the data (albeit the primary analysis is post-intervention), and also be better equipped to handle missing data without the need for imputation, as well as account/test for mechanisms that may bias the outcome (e.g. gender)? (see the extensive work by Jos Twisk and colleagues on GEE and random coefficient analysis, handling missing data etc.) Please clarify. Page 14; Safety  - It would be great to add a line or two to explain the safety measures in place during the exercise sessions. This may facilitate knowledge translation to other settings where minimal equipment is available (e.g. medical professional on site, availability of oxygen etc). Minor comments Title  - Please consider a different abbreviation; the “MISTER” feels (though I understand this is personal) a bit “gender biased”. Abstract  - Please consider adding the number of participants to be included. Page 9, line 28  - Please remove extra dot. Page 19 , line 20  - Typo; Novartis.
--	---

REVIEWER	Cheng, Sonia The University of Sydney
REVIEW RETURNED	16-Jan-2021

GENERAL COMMENTS

This study protocol is well-written with sound methodology, and puts forth a persuasive rationale for the potential benefits of minimal equipment PR programs. I commend the authors on undertaking this ambitious project. I have a few suggestions to improve the manuscript.

Abstract:

Page 4, line 20: The primary outcome (exercise capacity) should be stated here, in place of 'health outcomes'. Will people of any chronic lung disease be eligible for recruitment; clinically stable vs following an acute exacerbation? This should be clear.

Page 4, line 26: Sample size for recruitment (362 participants) should be stated.

Page 4, line 41: The type of analysis used to evaluate between-group differences (independent t-tests) should be stated.

Introduction:

Page 6, line 40: A clear definition of minimal equipment PR programs would be useful here with respect to setting and level of supervision, to distinguish such programs from home-based PR.

Page 8, line 3: I do not think that the study by Holland et al can be used to support minimal equipment PR, as the methodology differs significantly (ie, home-based rather than centre- or community-based; partial supervised vs fully supervised).

Methods:

Page 11, line 1: Randomisation should also be stratified by exercise capacity, given the primary outcome is ISWD and a mix of patients with chronic lung disease will be recruited.

Page 13, line 19: Will six-minute walk distance or endurance shuttle walk distance also be included as outcome measures, which may be more responsive to changes in submaximal exercise capacity? This was indicated in the systematic review by Alison et al, which indicated clinically significant improvements in 6MWD and ESWD following minimal equipment PR programs, but not ISWD.

Page 13, line 38: A more appropriate outcome measures of symptoms/dyspnoea may be CAT score or MMRC dyspnoea score, rather than the dyspnoea domain of the CRQ.

Page 14, line 16: The MCID for the ISWD suggested by the authors was established in COPD, but the present study will recruit any chronic lung disease including ILD and chest wall disease. Is a non-inferiority margin of 24 m therefore appropriate for the mixed chronic lung disease sample?

Page 14, line 44: With evaluating between-group differences, will the post-PR scores of the PR-min and PR-gym groups be compared at eight weeks, or will the change in score from baseline to eight weeks be compared between the PR-min and PR-gym groups? Please clarify so the text is consistent with the outcomes stated in Figure 1.

	Page 14, line 52: Clinical trials evaluating novel modes of exercise training with 3 timepoints (eg, Wootton 2014 Eur Respir J, Holland 2016 Thorax) have analysed differences between groups for change over time with linear mixed models or with ANCOVA (with baseline values as the covariate). These methods of data analysis may be more appropriate than independent sample t-tests. Page 14, line 54: It is unclear why the significance level has been set at 0.025, rather than the conventional 0.05. Page 15, line 6: Defining adherence as attending 8 of 16 sessions (50%) is relatively low and inflexible, in my opinion. A systematic review (Williams 2014 Respirology) defined program completion as undertaking 70% of planned sessions. Page 15, line 26: Could the authors provide additional detail on complier average causal effect (CACE) analysis and how this will be undertaken? It is unclear what “if non-compliance with is >10%” refers to. Discussion: Page 18, line 16: This sentence is complete: “One observational study demonstrated that PR-min is non-inferior to PR-gym in terms of exercise capacity and health-related quality of life but indicating that further investigation using a randomised controlled trial is...”
--	---

REVIEWER	Cox, Narelle Monash University , Allergy, Clinical Immunology and Respiratory Medicine
REVIEW RETURNED	19-Jan-2021

GENERAL COMMENTS	Thank you for the opportunity to review this study protocol relating to the MISTER trial – a randomised non-inferiority comparison of pulmonary rehabilitation using specialist gym equipment to a program using minimal/portable equipment. This work has the potential to improve access to effective, evidence-based alternative models of pulmonary rehabilitation. The protocol is well written and provides a clear overview of the study processes. The recruitment targets are substantial and I wish the authors the best of luck with their work. I have one minor point for the authors to address: - could the authors please specify the study dates i.e. when recruitment started and/or stopped and current trial status eg. recruiting/completed as per BMJ Open requirements for a protocol I have a couple of minor points the authors may wish to consider: - Page 9 Line 59 (Study Intervention) indicates that ‘initial endurance cycling speed will be set at 60% peak workload on a cycle ergometer’. Might the authors like to specify the manner/protocol in which this workload will be determined/tested? Only the ISWT is listed as a test of exercise capacity in the outcomes. - Page 13 Line 26 (Statistical analysis): is there a word(s) missing in the sentence ‘If non-compliance with is >10% a Complier-Average Causal-Effect...’ ? non-compliance with the intervention is >10%
--

	- Page 16 Line 16-21 (Discussion): This sentence appears incomplete. Perhaps 'required' at the end of the sentence? - Page 16 Line 44 (Discussion): This may just be semantics, but might the authors consider clarifying again here in the discussion that the 12-month follow-up is 12-months from time of recruitment, as opposed to 12-months after the intervention.
--	---

VERSION 1 – AUTHOR RESPONSE

Reviewer 1 comments:

C5: “Page 7; Methods: There are no timelines provided for the study (e.g. start of recruitment etc, and anticipated duration of the study). Please include..”

R5: Thank you for spotting this. We have added the study timeline to the methods section.

C6: “Page 9 – 11; Interventions: While I understand this is unlikely at this phase of the study, it would have been great to have a third arm in which PR-min was provided virtually / remotely, to an home-based environment. This would expand the external validity of the study to settings where also the physical resources, human-resources, and/or personal context does not facilitate outpatient visits (beyond the implications of COVID-19).”

R6: Thank you for this feedback. We agree that studying virtual, remote home-based PR is interesting, but it introduces different factors that might influence the efficacy of the pulmonary rehabilitation intervention such as the role of technology, home environment, digital literacy, degree of supervision etc. The current study will answer the question as to whether providing outpatient centre-based PR without specialist exercise equipment is non-inferior to outpatient centre-based PR with access to specialist exercise equipment.

C7: “Please add detail on any interaction between participant and team during, or in preparation of, the follow-up period. Are those in PR-min provided with the tools to continue their program, are patients in PR-gym directed to appropriate facilities in their home environment? Will there be phone follow-ups, booster sessions etc?”

R7: Thank you for this question. During the pulmonary rehabilitation programme, participants in both arms of the trial are encouraged to perform at least one additional exercise session at home (this is described in the methods section). In addition, as per national quality standards, participants in both arms of the trial are provided with an individualised, structured, written exercise plan for ongoing exercise maintenance. We have added the provision of the written exercise plan to the methods section. In order to reflect usual clinical care, neither trial arms will receive telephone or in-person contact, or booster sessions in the time-period between Visits 2 (post-pulmonary rehabilitation assessment) and 3 (12-month assessment).

C8: “The “education component” should be expanded upon, as this comprises a sizeable portion of the intervention and could be an important driver for changes in health-related quality of life (directly), as well as exercise-related outcomes (indirectly, through promotion of disease understanding and continued exercise during follow-up phase specifically).”

R8: Thank you for this feedback. As we are mindful of the word count limit, we have described the formal education component of pulmonary rehabilitation in the online supplement.

C9: “Methods: The main elements of the home exercise manual are briefly described in the appendix, but this is not a full manual that would easily enable replication. Would it be possible to state whether the exercise manual could be provided on application to the authors, or is available in full elsewhere?
R9: Thank you. The home exercise manual is the copyright of the Royal Brompton and Harefield Hospitals. We have amended the online supplement, stating that the home-exercise manual can be provided on application to the hospital.

C10: “Page 10, line 10 - Please consider adding these SOPs (progression schedule) as appendix.”
R10: Thank you. The standard operating procedures are the copyright of the Royal Brompton and Harefield Hospitals. They can be provided on application to the Royal Brompton and Harefield Hospitals; this has been added to the online supplement.

C11: “Page 12, statistical analysis - Would a more contemporary analysis like multi-level or GEE not be more appropriate given the longitudinal nature of the data (albeit the primary analysis is post-intervention), and also be better equipped to handle missing data without the need for imputation, as well as account/test for mechanisms that may bias the outcome (e.g. gender)? (see the extensive work by Jos Twisk and colleagues on GEE and random coefficient analysis, handling missing data etc.) Please clarify.”

R11: Thank you for your suggestions. We have planned to use the simple two-sample t-test for three main reasons: 1) our primary outcome was the change in exercise capacity from visit 1 to visit 2, which has no violation of assumption of independence. Therefore, there is no imperative reason to use the models (e.g. GEE, LMM) accounting for correlated data (Hubbard et al Epidemiology 2010); 2) the sample size was planned based on the two sample t-test, as although complex models have the advantages in dealing with covariates and missing data, they also involve more assumptions, such as the relationships among variables, correlation matrix. In our case, there were no prior data on which we can base to state and justify those assumptions. Furthermore, GEE and LMM usually require a large size of samples to make robust statistical inferences. In fact, for small, correlated samples, GEE and LMM did not show apparent superiority over the two-simple t-test (Huang et al Ophthalmic Epidemiology 2018); 3) the patients will be randomized into two groups via minimization on three key factors (previous status of PR completion, frailty status and deprivation). The minimization will ensure that these key factors are balanced between the two groups. Therefore, we did not plan to adjust for imbalance at the baseline and felt that the two-sample t-test is adequate. However, we take your point, we have added a GEE based modelling analysis as a sensitivity analysis.

C12: “Page 14; Safety - It would be great to add a line or two to explain the safety measures in place during the exercise sessions. This may facilitate knowledge translation to other settings where minimal equipment is available (e.g. medical professional on site, availability of oxygen etc).”

R12: Thank you for this feedback. Owing to word count constraints, we have added information on safety measures for PR-min and PR-gym to the online supplement.

C13: “Title - Please consider a different abbreviation; the “MISTER” feels (though I understand this is personal) a bit “gender biased”. ”

R13: Thank you for this feedback. It was never our intention that the study acronym would insinuate that the study was gender biased. As you’re aware, the inclusion criteria state that both men and women will be approached to participate in the study. Neither of the patient and public representatives (both women) on the study management group nor members of the research teams (majority women) that helped design the study and choose the acronym raised concerns about gender bias. In addition, neither the funding body nor ethics committee raised this issue. At the current time, the study has already commenced recruitment and the patient-facing and regulatory paperwork already established with the “MISTER” acronym – it is therefore too late to change the study acronym. However, we will remove the MISTER acronym from the title of the manuscript.

C14: "Abstract - Please consider adding the number of participants to be included."

R14: Thank you, we have amended the abstract.

C15: "Page 9, line 28 - Please remove extra dot."

R15: Thank you, we have removed it.

C16: "Page 19, line 20 - Typo; Novartis."

R16: Thank you, we have corrected this error.

Reviewer 2 comments:

C17: "Abstract: Page 4, line 20: The primary outcome (exercise capacity) should be stated here, in place of 'health outcomes'. Will people of any chronic lung disease be eligible for recruitment; clinically stable vs following an acute exacerbation? This should be clear."

R17: Thank you, we have added exercise capacity to the sentence.

C18: "Abstract: Page 4, line 26: Sample size for recruitment (362 participants) should be stated."

R18: Thank you, we have added this information.

C19: "Abstract: Page 4, line 41: The type of analysis used to evaluate between-group differences (independent t-tests) should be stated."

R19: Thank you. Please see R11 to C11.

C20: "Introduction: Page 6, line 40: A clear definition of minimal equipment PR programs would be useful here with respect to setting and level of supervision, to distinguish such programs from home-based PR."

R20: Thank you for this comment. We have amended the description of PR-min on page 4 of the introduction section (where it is first described) to reflect your feedback.

C21: "Introduction: Page 8, line 3: I do not think that the study by Holland et al can be used to support minimal equipment PR, as the methodology differs significantly (ie, home-based rather than centre- or community-based; partial supervised vs fully supervised)."

R21: Thank you for this comment. We agree and have adjusted the introduction to reflect the reviewer's comment.

C22: "Methods: Page 11, line 1: Randomisation should also be stratified by exercise capacity, given the primary outcome is ISWD and a mix of patients with chronic lung disease will be recruited."

R22: Thank you for this comment. We decided not to include baseline exercise capacity as a minimisation criterion because there is conflicting data as to whether it is associated with improvement in exercise capacity. One of the minimisation criteria, frailty, is associated with low exercise capacity (Maddocks et al Thorax 2016), and given the sample size, the randomisation procedure should minimise imbalance in exercise capacity across the two groups.

C23: "Methods: Page 13, line 19: Will six-minute walk distance or endurance shuttle walk distance also be included as outcome measures, which may be more responsive to changes in submaximal exercise capacity? This was indicated in the systematic review by Alison et al, which indicated clinically significant improvements in 6MWD and ESWD following minimal equipment PR programs, but not ISWD."

R23: Thank you for this comment. We will measure exercise capacity using the incremental shuttle walk test (ISW) only to reflect the clinical practice of the study centre, and to reduce burden upon patients. We were also keen to particularly target the ISW as it is considered a maximal test of exercise capacity, unlike the ESW and the 6MWD which are considered sub-maximal tests. We have also demonstrated that the ISW is responsive to both PR-min and PR-gym in a large propensity matched analysis of these two types of rehabilitation programmes (Patel et al Thorax 2021).

C24: "Methods: Page 13, line 38: A more appropriate outcome measures of symptoms/dyspnoea may be CAT score or MMRC dyspnoea score, rather than the dyspnoea domain of the CRQ."

R24: Thank you for this feedback. However, we respectfully disagree that the CAT or mMRC would be a more suitable measure of dyspnoea than the CRQ-dyspnoea domain. The CAT is a measure of health status rather than dyspnoea. In addition, as it is designed for people with COPD it would not be appropriate to use it in this study which aims to recruit people with chronic lung disease. Although useful as stratification tools, the mMRC nor MRC are measures of respiratory disability, not dyspnoea. They produce discrete values and are not particularly sensitive to change. As this study aims to compare the responsiveness of two interventions, we felt that it wasn't the most appropriate outcome measure to detect change in dyspnoea. We decided to use the CRQ-dyspnoea domain as it is an accepted, reliable and valid measure of breathlessness that is responsive to pulmonary rehabilitation (Williams et al Thorax 2001, Williams et al Thorax 2003, Schunemann et al COPD 2005, Schunemann et al ERJ 2005, Vodanovich et al Respiration 2015). Furthermore, we have demonstrated that it is responsive to both PR-min and PR-gym in a propensity matched analysis of these two types of rehabilitation programmes (Patel et al Thorax 2021).

C25: "Methods: Page 14, line 16: The MCID for the ISWD suggested by the authors was established in COPD, but the present study will recruit any chronic lung disease including ILD and chest wall disease. Is a non-inferiority margin of 24 m therefore appropriate for the mixed chronic lung disease sample?"

R25: Thank you. The reviewer raises a good point, but given that COPD is by far the largest patient group undergoing pulmonary rehabilitation, and the data for minimum clinically important difference of the ISW is best established in patients with COPD, determining the non-inferiority margin according to COPD was considered to be the most reasonable decision by the trial management group. The MCID of the ISW for patients with IPF and bronchiectasis are very similar to that seen in COPD (Nolan et al Thorax 2018, Walsh et al Annals of ATS 2020). For other conditions (e.g. chest wall disease) that might get referred for pulmonary rehabilitation, the MCID for the ISW is not established, but we would expect them to comprise a minority of the trial population.

C26: "Methods: Page 14, line 44: With evaluating between-group differences, will the post-PR scores of the PR-min and PR-gym groups be compared at eight weeks, or will the change in score from baseline to eight weeks be compared between the PR-min and PR-gym groups? Please clarify so the text is consistent with the outcomes stated in Figure 1."

R26: Thank you. Change in scores from baseline to eight weeks or 12 months will be compared between the intervention and control groups. We have clarified this in the text.

C27: "Methods: Page 14, line 52: Clinical trials evaluating novel modes of exercise training with 3 timepoints (eg, Wootton 2014 Eur Respir J, Holland 2016 Thorax) have analysed differences between

groups for change over time with linear mixed models or with ANCOVA (with baseline values as the covariate). These methods of data analysis may be more appropriate than independent sample t-tests.”

R27: Thank you. Please refer to our earlier response to C11.

C28: “Methods: Page 14, line 54: It is unclear why the significance level has been set at 0.025, rather than the conventional 0.05.”

R28: This is an inferiority trial. Unlike superiority trials, inferiority trials concern about the one-side comparison only (Dunn et al Trials 2018).

C29: “Methods: Page 15, line 6: Defining adherence as attending 8 of 16 sessions (50%) is relatively low and inflexible, in my opinion. A systematic review (Williams 2014 Respiriology) defined program completion as undertaking 70% of planned sessions.”

R29: Thank you for this feedback. There is no consensus, either in academic or clinical settings, on the number of sessions that defines pulmonary rehabilitation completion. For example, the National Asthma and COPD Audit Programme in the UK defines completion as attendance to an end of programme assessment without reference to the actual number of sessions attended.

The definition of 70% attendance used by Williams et al was completely arbitrary. A percentage of planned sessions (as suggested by the reviewer) might lead to a nonsensical paradox whereby someone that has completed at least 70% of 12 planned sessions (eg 9 sessions) would be considered a completer whereas someone who has attended 50% of 24 planned sessions (eg 12 sessions) and therefore attended numerally more supervised sessions, would be classified as a non-completer.

We based our value of eight sessions on local practice of the trial centre. Evidence to support this comes from the study by Sewell and colleagues that demonstrated that a twice-weekly, four-week pulmonary rehabilitation programme (eight sessions) was associated with similar outcomes as a twice-weekly, seven-week programme (14 sessions) (Sewell et al Thorax 2006).

C30: “Methods: Page 15, line 26: Could the authors provide additional detail on complier average causal effect (CACE) analysis and how this will be undertaken? It is unclear what “if non-compliance with is >10%” refers to.”

R30: The CACE analysis will be applied (as one of the sensitivity analyses) if the non-compliance rate is >10%. The CACE analysis is a subgroup analysis of the average intervention effects among all participants in the PR-min group compared with those in the PR-gym group who would have been compliant with the PR regimen had they been randomised to the PR-min. It is done through an instrumental variable – the proportion of PR-min participants found to be compliant (Peugh et al J School Psychology 2017).

C31: “Discussion: Page 18, line 16: This sentence is complete: “One observational study demonstrated that PR-min is non-inferior to PR-gym in terms of exercise capacity and health-related quality of life but indicating that further investigation using a randomised controlled trial is...”

R31: Thank you for spotting this error, we have corrected this.

Reviewer 3 comments:

C32: "Thank you for the opportunity to review this study protocol relating to the MISTER trial – a randomised non-inferiority comparison of pulmonary rehabilitation using specialist gym equipment to a program using minimal/portable equipment. This work has the potential to improve access to effective, evidence-based alternative models of pulmonary rehabilitation. The protocol is well written and provides a clear overview of the study processes. The recruitment targets are substantial, and I wish the authors the best of luck with their work."

R32: Thank you for this feedback.

C33: "I have one minor point for the authors to address: - could the authors please specify the study dates i.e. when recruitment started and/or stopped and current trial status e.g. recruiting/completed as per BMJ Open requirements for a protocol."

R33: Thank you, this information has been added to the methods section.

C34: "Page 9 Line 59 (Study Intervention) indicates that 'initial endurance cycling speed will be set at 60% peak workload on a cycle ergometer'. Might the authors like to specify the manner/protocol in which this workload will be determined/tested? Only the ISWT is listed as a test of exercise capacity in the outcomes."

R34: Thank you for spotting this error, we have corrected this.

C35: "Page 13 Line 26 (Statistical analysis): is there a word(s) missing in the sentence 'If non-compliance with is >10% a Complier-Average Causal-Effect...' ? non-compliance with the intervention is >10%".

R35: Thank you for spotting this. We have corrected the manuscript.

C36: "Page 16 Line 16-21 (Discussion): This sentence appears incomplete. Perhaps 'required' at the end of the sentence?"

R36: Thank you for spotting this error, we have corrected this.

C37: "Page 16 Line 44 (Discussion): This may just be semantics, but might the authors consider clarifying again here in the discussion that the 12-month follow-up is 12-months from time of recruitment, as opposed to 12-months after the intervention."

R37: Thank you, we have amended the manuscript.

Thank you very much for considering the revised manuscript for publication in BMJ Open. We hope that we have answered all the reviewers' comments satisfactorily and would be very happy to provide

any further clarification, comments or changes as necessary. Owing to the changes requested by the reviewers, the word count has increased to 4128 words.

VERSION 2 – REVIEW

REVIEWER	Heine, Martin Stellenbosch University, Institute of Sports and Exercise Medicine
REVIEW RETURNED	04-Jun-2021

GENERAL COMMENTS	Dear Dr. Nolan and colleagues, Thank you for considering the suggested changes / queries. I wish you all the best in this huge yet important undertaking, nb. in challenging times.
--

REVIEWER	Cox, Narelle Monash University , Allergy, Clinical Immunology and Respiratory Medicine
REVIEW RETURNED	02-Jun-2021

GENERAL COMMENTS	Thank you to the authors for their comprehensive response to reviewers. I have no further comments to make regarding the manuscript. I think there may be a typographical error in the methods indicating study timelines, that the final V2 is anticipated to occur 31/05/2022, not 2023 as written. The authors may wish to confirm that the dates in the manuscript correspond with the trial registration.
---